# Clinical Efficacy of Hypofractionated Proton Beam Therapy for Intrahepatic Cholangiocarcinoma

**DOI:** 10.3390/cancers14225561

**Published:** 2022-11-12

**Authors:** Tae Hyun Kim, Sang Myung Woo, Woo Jin Lee, Jung Won Chun, Yu Ri Cho, Bo Hyun Kim, Young-Hwan Koh, Sang Soo Kim, Eun Sang Oh, Do Yeul Lee, Sung Uk Lee, Yang-Gun Suh, Sung Ho Moon, Joong-Won Park

**Affiliations:** 1Center for Proton Therapy, National Cancer Center, Goyang 10408, Korea; 2Center for Liver and Pancreatobiliary Cancer, National Cancer Center, Goyang 10408, Korea

**Keywords:** intrahepatic cholangiocarcinoma, overall survival, proton beam therapy, radiotherapy, freedom from local progression

## Abstract

**Simple Summary:**

Proton beam therapy (PBT) has the potential to improve local tumor control and subsequently improve survival of patients with inoperable or recurrent intrahepatic cholangiocarcinoma (IHCC); however, studies on PBT in patients with IHCC are still limited. This study evaluated the efficacy and safety of hypofractionated PBT in IHCC patients with inoperable or recurrent disease. Our findings demonstrated that hypofractionated PBT was considered tolerable and safe and offers a high rate of local tumor control and promising survival in patients with IHCC. In addition, the survival outcomes in selected patients with localized disease treated with hypofractionated PBT were comparable to those of surgical resection. Further large-scale studies are warranted to confirm these findings.

**Abstract:**

Forty-seven patients with intrahepatic cholangiocarcinoma (IHCC) who received proton beam therapy (PBT) were analyzed to evaluate the clinical efficacy and safety of hypofractionated PBT in patients with inoperable or recurrent IHCC. The median prescribed dose of PBT was 63.3 GyE (range: 45–80 GyE) in 10 fractions, and the median duration of follow-up in all the patients was 18.3 months (range: 2.4–89.9 months). Disease progression occurred in 35 of the 47 (74.5%) patients; local, intrahepatic, and extrahepatic progression occurred in 5 (10.6%), 20 (42.6%), and 29 (61.7%) patients, respectively. The 2-year freedom from local progression (FFLP), progression-free survival (PFS), overall survival (OS) rates, and median time of OS were 86.9% (95% confidence interval [CI], 74.4–99.4%), 16.8% (95% CI, 4.3–29.3%), 42.7% (95% CI, 28.0–57.4%), and 21.9 months (95% CI, 16.2–28.3 months), respectively; grade ≥ 3 adverse events were observed in four (8.5%) patients. In selected patients with localized disease (no viable tumors outside of the PBT sites), the median time of OS was 33.8 months (95% CI, 5.4–62.3). These findings suggest that hypofractionated PBT is safe and could offer a high rate of FFLP and promising OS in patients with inoperable or recurrent IHCC.

## 1. Introduction

Intrahepatic cholangiocarcinoma (IHCC) is the second most common tumor arising from the epithelium of the bile ducts within the liver (followed by hepatocellular carcinoma) and accounts for approximately 10% of primary liver cancers. Surgery is generally considered the only curative treatment for IHCC. Unfortunately, only 10–30% of the patients are candidates for surgery at the time of diagnosis, and half of the patients undergoing surgery experience recurrence within 1 year, even after adjuvant treatment [1]. For patients with inoperable or recurrent IHCC, chemotherapy with combinations of multiple agents has demonstrated modest prolongation of survival; however, this is still far from a cure [2,3,4], and local progression is common [5,6]. Thus, liver-directed local treatments, including radiotherapy (RT) and arterially directed therapies, are considered for achieving local tumor control.

Several previous studies have demonstrated RT to improve local tumor control and prolong survival in patients with inoperable or recurrent IHCC [5,7,8,9,10,11]. Recent technical advances in RT planning and delivery, including intensity-modulated RT (IMRT), stereotactic body RT (SBRT), respiratory gating techniques, image guidance techniques using computed tomography (CT) or magnetic resonance imaging (MRI), and proton beam therapy (PBT), have enabled the safe delivery of high radiation doses to tumor(s) and have been used for improving local tumor control and survival [6,12,13,14,15,16,17,18,19,20,21,22,23,24]. With the advantage of physical characteristics of the proton beam (known as ‘Bragg peak’), hypofractionated PBT can improve the therapeutic ratio by escalating the delivered radiation doses to the tumor(s) while minimizing the delivered radiation doses to the surrounding normal tissues; moreover, PBT has been proven to be an effective treatment for primary and secondary liver tumors [19,25,26,27,28,29]. To date, studies on PBT in patients with IHCC are limited owing to the rarity of IHCC [19,20,21,22]. However, hypofractionated PBT may have several merits, including potential improvement in the therapeutic ratio, reduction of the overall treatment time, and reduction in the number of chemotherapy breaks. Based on this background, hypofractionated PBT with/without various schedules and regimens of chemotherapy, based on each patient’s age and various medical conditions, was performed in patients with inoperable or recurrent IHCC at our institution. This study aimed to evaluate the clinical efficacy and safety of hypofractionated PBT in these patients.

## 2. Materials and Methods

### 2.1. Patients

Patients treated with PBT for primary or recurrent IHCC between May 2012 and July 2021 were registered in a database and reviewed. Patients aged 18 years or older with histologically confirmed IHCC who received PBT for intrahepatic lesions(s) were included in this study. Patients who underwent PBT for lesion(s) not located in the liver were excluded from this study. All the patients underwent multidisciplinary evaluation by medical, surgical, and radiation oncologists for assessment of resectability and selection of treatment modalities; i.e., the use, sequence, and regimens of systemic treatments and the use of local treatments (including PBT), considering the extent of disease and each patient’s performance status (PS) and underlying medical conditions, and were staged according to the American Joint Committee on Cancer (AJCC) staging classifications [30]. Demographic and clinical data were collected from medical records, including sex, age, Eastern Cooperative Oncology Group (ECOG) PS, tumor size and stage, carbohydrate antigen (CA) 19–9, hilum invasion, history of previous biliary drainage, treatments before, during, and after PBT, prescribed radiation dose of PBT, disease progression sites and times, etc.; these data were anonymized after assigning case numbers. All the methods in the present study were conducted in compliance with the relevant regulations and guidelines, including the Declaration of Helsinki and Good Clinical Practice guidelines. The Institutional Review Board of the NCC approved the present study (NCC20220080) and waived the requirement for written informed consent owing to the retrospective design of the present study.

### 2.2. Treatment

The PBT procedures have been described in detail in previous reports [26,27,31,32,33,34,35]. Patients were immobilized in the supine position with a customized vacuum cushion and underwent contrast-enhanced four-dimensional CT imaging under gating of respiration. The gross tumor volume (GTV) was defined in average intensity projection CT images at exhalation (gated) phases (i.e., 30% of the entire respiration cycle in each patient) based on available imaging studies, including liver dynamic CT and/or MRI. The internal target volume (ITV) and organs at risk (OARs) were defined to account for the extent and movement of the GTV and each OAR at the exhalation (gated) phases, and the clinical target volume was defined as the ITV with no additional margins [25,27,31,32,34,36]. The planning target volume (PTV) was defined as the ITV plus a 5–7 mm margin in all directions. PBT (Eclipse; Version 13.7, Varian Medical System, Palo Alto, CA, USA) was performed using 3 (range, 2–4) 230 MeV double-scattered proton beams (Proteus 235; Ion Beam Applications, S.A., Louvain-la-Neuve, Belgium), with the intention that 100% of the prescribed dose covered at least 95% of the PTV (Appendix A). The radiation dose of PBT was expressed as Gray equivalent (GyE = physical dose of proton beam [Gray] × relative biologic equivalent factor of proton beam [1.1]) and was converted to the equivalent dose for 2 Gy fractions (EQD2 [GyE_3_ or GyE_10_] = [(fraction dose + α/β)/(2 + α/β)] × total dose) with α/β values of 10 (acute responding tissues and tumor) or 3 (late responding tissues) (GyE_10_ or GyE_3_, respectively) [37]. Median prescribed dose and EQD2 of PBT were 63.3 GyE (range, 45–80 GyE) in 10 fractions and 86.2 GyE_10_ (range, 54.4–120 GyE_10_), respectively. The prescribed dose was dependent on the tumor location; 66–80 GyE in 10 fractions (91.3–120 GyE_10_) was prescribed for patients with tumor(s) located more than 2 cm from the gastrointestinal (GI) organs that did not directly contact the hepatic hilum area, and 45–60 GyE in 10 fractions (54.4–80 GyE_10_) was prescribed for patients with tumor(s) located within 2 cm from the GI organs that directly contacted the hepatic hilum area, while maintaining the dose-volume constraints for the OARs. The details of the dose-volume constraints for the OARs have been described in previous reports [27,28,31,32,33]. The radiation doses to the esophagus, stomach, and bowel (duodenum, small and large intestine) of 2 cm^3^ were less than 39 GyE, 37 GyE, and 35 GyE, respectively, the maximum doses to the spinal cord were less than 39 GyE, and the irradiated relative volumes of the remaining residual liver (total liver–GTV) and total liver receiving more than 27 GyE were less than 50% and 60%, respectively. All the patients were instructed to fast for at least 4 h prior to PBT to minimize inter-fractional uncertainty, and irradiation was performed during the gated phases after localization using digital orthogonal and/or cone beam CT images.

### 2.3. Assessments and Statistical Analysis

Patients were assessed weekly during PBT and, after the completion of PBT, they were assessed at the first month, every three months for the first two years and every six months thereafter. Clinical, laboratory, and imaging tests, including abdominal CT and/or MRI, were performed at each visit. Local, intrahepatic, and extrahepatic progression were defined as the presence of a growth or new tumor within 1 cm from the margin of the PTV, within the liver, and beyond the 1 cm margin of the PTV and the liver (including the regional or non-regional lymph nodes and distant organs), respectively, according to the Response Evaluation Criteria in Solid Tumors (version 1.1) [38]. Adverse events (AEs) were assessed using the Common Terminology Criteria for AEs (version 5.0). Times for freedom from local progression (FFLP), progression-free survival (PFS), and overall survival (OS) were defined as the interval from the commencement date of PBT to the date of local progression, disease progression or death, and death or the last follow-up, respectively. Comparisons of the categorical and continuous variables were performed using Fisher’s exact test and the t-test, respectively, and the probability of survival was estimated using the Kaplan–Meier method. The log-rank test was used to compare the survival differences in the univariate analysis, and in the multivariate analysis, a stepwise forward selection procedure containing the variables with univariate statistical significance of *p* < 0.1 was used. The hazard ratios (HRs) were estimated using the Cox proportional hazards model. Statistical significance was set at *p* < 0.05, and STATA software (version 14.0; StataCorp, College Station, TX, USA) was used for all the statistical analyses.

## 3. Results

Fifty-one patients with primary or recurrent IHCC underwent PBT between May 2012 and July 2021. Among them, except for four patients who received PBT for recurrent extrahepatic lesions, 47 patients who received PBT for intrahepatic lesions were included in this study. None of the patients were candidates for surgical resection based on a multidisciplinary evaluation, and the patient characteristics are shown in Table 1. Of the 47 patients, all the viable tumor burden(s) could be irradiated by PBT in 29 (61.7%) patients, and all the viable tumor burden(s) (including metastatic lesion(s) at lymph nodes and distant organs) could not be sufficiently irradiated by PBT in 18 (38.3%) patients. Patients with viable tumor(s) outside of the PBT site(s) had significantly higher frequencies of N+, M1, and advanced stage (III–IV) (*p* < 0.05 each) and larger trends in tumor size and PTV than patients with no viable tumor(s) outside of the PBT site(s) (*p* < 0.05 each) (Table 1). Thirty-eight patients had newly diagnosed disease, and 9 patients had recurrent disease following surgical resection. Most (n = 34, 72.3%) of the patients had IHCC with no hilum invasion, and the remaining 13 (27.7%) patients had IHCC with hilum invasion; six (12.8%) patients had no biliary drainage and seven (14.9%) patients had biliary drainage. Prior to PBT, 21 (44.7%) patients received treatment (including chemotherapy) (Appendix A). During PBT, concurrent chemotherapy was considered in 39 patients, except for 8 patients with stage I/II, but 10 (25.6%) patients received concurrent chemotherapy by the physicians’ decision considering the patient’s PS and age (Table 1). The median EQD2 of PBT was 80 GyE_10_ (range, 54.4–120.0); 35 (74.5%) patients received >80 GyE_10_, and 12 (25.5%) patients received ≤80 GyE_10_. After PBT, subsequent treatments (including chemotherapy) were considered for all the patients based on their PS and age, and 24 (51.1%) patients received subsequent treatments (Appendix A).

The median follow-up duration of all the patients was 18.3 months (range, 2.4–89.9 months). At the time of analysis, 33 of 47 (70.2%) patients died from disease progression (n = 29) and unknown causes (n = 4), and 35 of 47 (74.5%) patients had disease progression. Local, intrahepatic, and extrahepatic progression were observed in 5 (10.6%), 20 (42.6%), and 29 (61.7%) patients (Figure 1).

The FFLP rates at 1-year and 2-year were 91.7% (95% confidence interval [CI], 82.7–100.7%) and 86.9% (95% CI, 74.4–99.4%), respectively, and the median time of FFLP was not reached (Appendix A). In the univariate and multivariate analyses, patients with T1-2, N0, M0, stages I–II, no hilum invasion, no biliary drainage, and EQD2 ≥ 80 GyE_10_ demonstrated a trend towards higher FFLP rates than those with T3-4, N+, M1, stages III and IV, biliary drainage, and EQD2 < 80 GyE_10_ with no statistical significance (*p* > 0.05 each) (Table 2 and Table 3) (Appendix A), and there were no significant differences in the FFLP rates between the two groups according to the absence or presence of viable tumor(s) outside of the PBT site(s) (Table 2) (Figure 2A).

The PFS rates at 1-year and 2-year were 43.4% (95% CI, 28.7–58.1%) and 16.8% (95% CI, 4.3–29.3%), respectively, and the median time of PFS was 8.3 months (95% CI, 2.8–13.7 months) (Appendix A). In the univariate analysis, patients with T1-2, N0, M0, stages I–II, history of pre-treatment, and no viable tumor(s) outside of the PBT site(s) had significantly higher PFS rates than those with T3-4, N+, M1, stages III and IV, no history of pre-treatment, and viable tumor(s) outside the PBT site(s) (*p* < 0.05 each) (Table 2) (Figure 2B). Patients with ECOG PS 0 and EQD2 of ≥ 80 GyE_10_ showed a trend toward higher PFS rates than those with ECOG PS 1–2 and EQD2 of < 80 GyE_10_ with no statistical significance (*p* > 0.05 each) (Table 2) (Appendix A). Only the status of viable tumor(s) outside of the PBT site(s) was significantly associated with PFS (*p* < 0.05) (Table 3).

The OS rates at 1-year and 2-year were 63.8% (95% CI, 50.1–77.5%) and 42.7% (95% CI, 28.0–57.4%), respectively, and the median OS was 21.9 months (95% CI, 16.2–28.3 months) (Appendix A). In the univariate analysis, patients with no viable tumor(s) outside of the PBT site(s) had significantly higher OS than those with viable tumor(s) outside of the PBT site(s) (median, 33.8 months [95% CI, 5.4–62.3%] vs. 7.6 [95% CI, 3.2–12.0%], *p* < 0.05) (Figure 2C), and ECOG PS, T, N, and M classification, stage, serum CA 19–9 level, and concurrent chemotherapy were also significantly associated with the OS (*p* < 0.05) (Table 2). Patients with EQD2 ≥80 GyE_10_ showed a trend towards higher OS rates than those with EQD2 <80 GyE_10_; however, the difference was not significant (*p* > 0.05) (Table 2) (Appendix A). The status of viable tumor(s) outside of the PBT site(s), ECOG PS, and serum CA 19–9 was significantly associated with the OS in the multivariate analysis (*p* < 0.05) (Table 3).

The AEs associated with PBT are summarized in Table 4. Hematologic AEs were observed in 20 (42.6%), six (12.8%), and four (8.5%) patients with grades 1, 2, and 3, respectively. Four grade 3 AEs (leukopenia [n = 2] and hyperbilirubinemia [n = 2]) were related to bone marrow suppression by chemotherapy and biliary obstruction by disease progression. Non-hematologic AEs were observed in 19 (40.4%) and three (6.4%) patients with grades 1 and 2, respectively. The most frequent non-hematologic AEs were skin reactions (grade 1, 15 [31.9%] patients; and grade 3, three [6.4%] patients) and radiation pneumonitis (grade 1, 12 [25.5%] patients). Grade 2 GI ulcer was observed in one (2.1%) patient, who recovered with medication. No PBT-related grade 4 AEs, hepatic failure, or death were observed.

## 4. Discussion

For patients with inoperable or recurrent IHCC, no standard treatment option for liver-directed local treatments has been established owing to the lack of randomized trials to elucidate their benefits. Several retrospective studies evaluating the role of conventional fractionated RT showed an improvement in the OS with RT over no treatment (median: 7–10 vs. 3–5 months, *p* < 0.05) [5,10] (Appendix A). These studies suggest that RT can improve the OS compared to no treatment or chemotherapy alone; however, the role of conventional fractionated RT with or without chemotherapy remains controversial owing to high rates of local progression [6]. Recently, hypofractionated RT with X-ray or SBRT with EQD2 of 25–150 GyE_10_ in 1–15 fractions has been attempted to improve local tumor control by delivering high RT doses to the tumor and reducing the duration of RT and breaks in chemotherapy and has demonstrated promising outcomes, including 2-year FFLP rates of 47–79%, median OS of 10–17 months, and grade 3 AEs of 9–19% [13,14,15,16,23,24] (Appendix A). Hypofractionated PBT with various EQD2 of 43.9–91.3 GyE_10_ in 10–37 fractions has also been performed in patients with IHCC and has demonstrated 2-year FFLP rates of 41.4–94.1%, median OS of 15–22.5 months, and grade ≥ 3 AEs of 0–17.3% [18,19,20,22] (Appendix A). Recently, Smart et al. [17] analyzed 66 patients with IHCC treated with hypofractionated RT with X-rays or proton beams, and patients treated with PBT demonstrated a trend towards higher OS than patients treated with RT with X-rays (*p* = 0.05) (Appendix A). In the present study, we applied hypofractionated PBT with a median EQD2 of 86.2 GyE_10_ (range, 54.4–120 GyE_10_) in 10 fractions, depending on the proximities and dose-volume constraints of OARs, and observed 2-year FFLP rates of 86.9%, median OS of 21.9 months, and grade ≥ 3 AE of 8.5%. Although directly comparing the results of the present study with those of previous studies is not possible owing to the different baseline and pre- and post-treatment characteristics, the FFLP and OS in the present study were at the higher end of the wide range previously reported [13,14,15,16,17,18,19,20,22,23,24] (Appendix A).

Several studies have suggested a dose-response relationship with OS in patients with IHCC. Tao et al. [12] analyzed 79 patients with IHCC treated with hypofractionated RT with X-ray or proton beams and showed that patients who received EQD2 > 67 GyE_10_ had better 3-year FFLP rates (78% vs. 45%, *p* < 0.05) and 3-year OS rates (73% vs. 38%, *p* < 0.05) than patients who received EQD2 > 67 GyE_10_. Makita et al. [20], in an analysis of 38 patients, including six patients with IHCC treated with PBT, reported that patients who received EQD2 ≥ 58.3 GyE_10_ had better 1-year OS (83.1% vs. 22.2%, *p* < 0.05) than those who received EQD2 < 58.3 GyE_10_. These results suggest that escalated RT doses can improve local tumor control and subsequently improve OS. In the present study, 47 patients with inoperable or recurrent IHCC were treated with PBT with a median EQD2 of 80 GyE_10_ (range, 54.4–120.0). Most of the patients (74.5%) received >80 GyE_10_, and the patients who received EQD2 ≥ 80 GyE_10_ had a trend towards higher 2-year FFLP (92.7% vs. 66.7%), PFS (11% vs. 4%), and OS (23.8% vs. 13.2%) than those who received EQD2 of <80 GyE_10_ with no statistical significance owing to the relatively small number of study populations (*p >* 0.05 each) (Table 2). In addition, 25.5% (12 or 47) of the patients had tumors close to the GI organs that received ≤80 GyE_10_ (range, 54.4–80 GyE_10_) within dose-volume constraints to the surrounding OARs, including the liver and GI organs; grade ≥ 3 AEs were observed in 8.5% (4 of 47) of the patients with no grade ≥ 3 GI AEs (Table 4). However, owing to the relatively small study population in previous studies and our study, further larger comprehensive studies are warranted to evaluate the potential risks of severe AEs when applying intensive RT and escalating the RT dose for patients with IHCC, especially in tumors close to the GI organs, large tumors, and a relatively small volume of residual liver (i.e., less than 700 cm^3^).

In patients with advanced biliary cancer, including IHCC with good PS, a randomized trial has demonstrated combination chemotherapy with gemcitabine and platinum to be superior to gemcitabine alone (11.7 vs. 8.1 months, *p* < 0.05) [4], and a prospective study with further intensive chemotherapy with gemcitabine, platinum, and nab-paclitaxel also has a promising median OS of 19 months [3]. However, the use and continuation of intensive chemotherapy in patients with IHCC is frequently limited by the high incidence (i.e., approximately 58–71%) of grade ≥ 3 AEs. In addition, local progression is one of common cause of disease progression [5,6]. Thus, RT with and without chemotherapy, depending on various clinical conditions of patients with inoperable or recurrent IHCC, has been attempted. Kim et al. [11] retrospectively analyzed 92 patients with IHCC, including 25 patients who received RT, with a mean RT dose of 44.7 Gy (range, 25–60 Gy), combined with chemotherapy, and 67 patients who received chemotherapy alone. The addition of RT to chemotherapy improved OS compared to chemotherapy alone (9.3 vs. 6.2 months, *p* < 0.05). Several studies have also suggested that the use of chemotherapy with RT and/or PBT can improve the OS [17,22]. Unfortunately, the incidence of IHCC was most frequent in the patients in their 70s, followed by those in their 80s and 60s [39]. The use of less intensive anti-cancer treatments, rather than intensive combination chemotherapy, has been considered in the real world owing to the high likelihood of patients being less likely to tolerate intensive chemotherapy with increasing age [17,20,21,22]. In the present study, approximately 40% of the patients were ≥ 70 years old, and approximately 50% of the patients did not receive chemotherapy prior to, during, or after PBT. Although the use of chemotherapy with PBT did not significantly improve the FFLP, PFS, and OS in the present study, further large-scale studies are warranted to evaluate the additional benefits of PBT over chemotherapy in patients with IHCC.

This study had several limitations. First, this study retrospectively analyzed a relatively small number of patients who received PBT with heterogeneous dose-fractionation schemes and the use and sequences of various chemotherapeutic regimens; thus, all potential biases were not thoroughly accounted for. However, the dose-fractionation schemes were determined according to the dose-volume constraints of OARs, including the liver and GI organs, and the use, sequences, and regimens of chemotherapy were decided by the physicians considering each patient’s age, PS, and concomitant medical conditions. In clinical practice, chemotherapy is administered first in patients who are expected to tolerate intensive chemotherapy, and other local treatments are considered first in patients who are expected to have difficulty tolerating intensive chemotherapy. In addition, local treatments are often applied in patients with IHCC with or without local progression on systemic therapy. Thus, the heterogeneity of the study population reflects real-world clinical practice. Second, treatment-related AEs could be underestimated in retrospective studies since they may not have been fully documented in the medical records. However, SBRT and PBT have been shown to have safe AE profiles [13,14,15,16,18,19,20,22,23,24]. In the present study, PBT showed good local tumor control and survival, i.e., 2-year FFLP rates of 86.9% and median OS of 21.9 months in patients with inoperable or recurrent IHCC with safe AE profile (i.e., grade ≥ 3 AE of 8.5% with no grade ≥ 3 GI AEs). In addition, PBT showed promising OS, that is, a median OS of 33.8 months, in patients with localized disease (no viable tumors outside of the PBT sites), comparable to those of surgical resection [40,41,42].

## 5. Conclusions

The present study demonstrated that hypofractionated PBT was tolerable and safe and could offer high rates of FFLP and promising OS in patients with inoperable or recurrent IHCC. In addition, hypofractinated PBT could result in comparable OS to surgical resection in selected patients with localized disease, that is, no viable tumors outside of the PBT sites. Although the need for high-cost dedicated equipment for PBT has limited clinical application of PBT for patients with IHCC to date, further prospective large-scale studies evaluating the benefits of PBT in these patients are warranted to verify these findings.

## Figures and Tables

**Figure 1 cancers-14-05561-f001:**
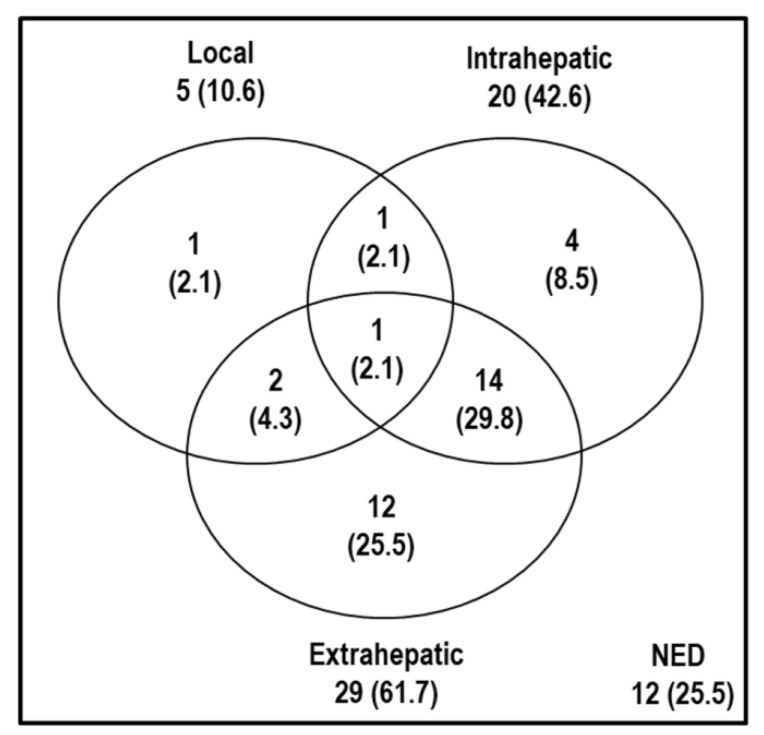
Patterns of disease progressions at the time of analysis. Abbreviations: Local, local progression; Intrahepatic, intrahepatic progression; Extrahepatic, extrahepatic progression; NED, no evidence of disease progression.

**Figure 2 cancers-14-05561-f002:**
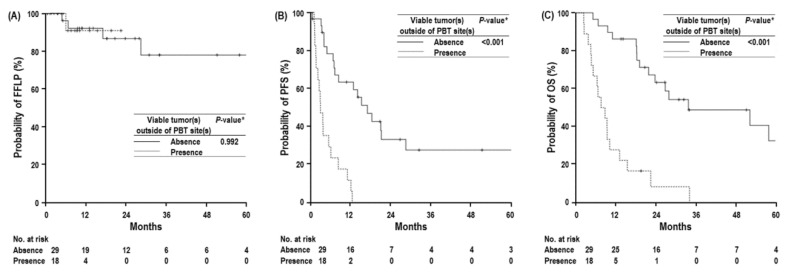
Freedom from local progression (FFLP) (**A**), progression-free survival (PFS) (**B**), and overall survival (OS) (**C**) 0 curves according to absence or presence of viable tumor(s) outside of PBT site(s). Abbreviations: PBT, proton beam therapy. * log-rank test.

**Table 1 cancers-14-05561-t001:** Patient and treatment characteristics.

			Viable Tumor(s) Outside of PBT Site(s)		
Characteristics		Total, n (%)	Absence, n (%)	Presence, n (%)	*p*-Value
Sex	Male	32 (68.1)	21 (72.4)	11 (61.1)	0.311 ^†^
	Female	15 (31.9)	8 (27.6)	7 (38.9)	
Age, years	Median (range)	67 (45–83)	62 (45–83)	69 (50–80)	0.426 ^‡^
	<70	28 (59.6)	18 (62.1)	10 (55.6)	0.444 ^†^
	≥70	19 (40.4)	11 (37.9)	8 (44.4)	
ECOG PS	0	28 (59.6)	21 (72.4)	7 (38.9)	0.041 ^†^
	1	18 (38.3)	8 (72.6)	10 (55.6)	
	2	1 (2.1)	0 (0.0)	1 (5.6)	
Hilum invasion	No	34 (72.3)	20 (69.0)	14 (77.8)	0.739 ^†^
	Yes	13 (27.7)	9 (31.0)	4 (22.2)	
Tumor size *, cm	Median (range)	5.2 (1.0–11.0)	4.8 (1.0–10.3)	5.7 (2.1–11.0)	0.070 ^‡^
	<5	21 (44.7)	16 (55.2)	5 (27.8)	0.061 ^†^
	≥5	26 (55.3)	13 (44.8)	13 (72.2)	
T classification	T1	3 (6.4)	3 (10.3)	0 (0.0)	0.360^†^
	T2	6 (12.8)	5 (17.2)	1 (5.6)	
	T3	34 (72.3)	19 (65.5)	15 (83.3)	
	T4	4 (8.5)	2 (6.9)	2 (11.1)	
N classification	N0	36 (76.6)	29 (100)	7 (38.9)	<0.001 ^†^
	N+	11 (23.4)	0 (0.0)	11 (61.1)	
M classification	M0	36 (76.6)	29 (100)	7 (38.9)	<0.001 ^†^
	M1	11 (23.4)	0 (0.0)	11 (38.3)	
Stage	I	3 (6.4)	3 (10.3)	0 (0.0)	<0.001 ^†^
	II	5 (10.6)	5 (17.2)	0 (0.0)	
	III	28 (59.6)	21 (72.4)	7 (38.9)	
	IV	11 (23.4)	0 (0.0)	11 (61.1)	
CA 19–9, U/mL	Median (range)	65.5 (2.0–6282.0)	35.5 (2.0–6282.0)	141.1 (5.0–5860.0)	0.980 ^‡^
	<100	27 (57.4)	18 (62.1)	9 (50.0)	0.304 ^†^
	≥100	20 (42.6)	11 (37.9)	9 (50.0)	
Biliary drainage	No	40 (85.1)	26 (89.7)	14 (77.8)	0.242 ^†^
	Yes	7 (14.9)	3 (10.3)	4 (22.2)	
Disease status	ND	38 (80.9)	22 (75.9)	16 (88.9)	0.239 ^†^
	IHR	9 (19.1)	7 (24.1)	2 (11.1)	
Pre-Tx to PBT site(s)	No	26 (55.3)	20 (69.0)	6 (33.3)	0.018 ^†^
	Yes	21 (44.7)	9 (31.0)	12 (66.7)	
Concurrent CTx ^§^	No	37 (78.7)	24 (82.8)	13 (72.2)	0.308 ^†^
	Yes	10 (21.3)	5 (17.2)	5 (27.8)	
Post-Tx after PBT	No	23 (48.9)	16 (55.2)	7 (38.9)	0.216 ^†^
	Yes	24 (51.1)	13 (44.8)	11 (61.1)	
Total dose	Median (range)	80.0 (54.4–120.0)	91.3 (54.4–120.0)	80.0 (62.5–99.2)	0.135 ^‡^
(EQD2, GyE_10_)	<80	12 (25.5)	5 (17.2)	7 (38.9)	0.096 ^†^
	≥80	35 (74.5)	24 (82.8)	11 (6.11)	
PTV, cm^3^	Median (range)	115.3 (9.7–1035.1)	76.9 (9.7–575.7)	142.8 (20.3–1035.1)	0.050 ^‡^
TL volume, mL	Median (range)	1251.4 (753.8–2557.7)	1245.4 (753.8–1984.7)	1315.9 (997.2–2557.7)	0.207 ^‡^
_TL_V_27GyE_, %	Median (range)	19.1 (2.0–63.7)	17.7 (2.0–49.5)	20.4 (3.9–63.7)	0.235 ^‡^
RRL volume, cm^3^	Median (range)	1150.8 (75.38–1914.0)	1157.2 (75.38–1914.0)	1148.7 (871.7–1148.7)	0.757 ^‡^
_RRL_V_27GyE_, %	Median (range)	13.8 (1.3–36.0)	13.0 (1.3–33.0)	13.9 (3.1–36.0)	0.407 ^‡^
_Stomach_D_2cc_, GyE	Median (range)	26.7 (0.0–36.4)	26.7 (0.0–36.4)	25.5 (0.0–35.6)	0.428 ^‡^
_Esophagus_D_2cc_, GyE	Median (range)	1.3 (0.0–40.6)	4.2 (0.0–37.5)	1.1 (0.0–40.6)	0.645 ^‡^
_Duodenum_D_2cc_, GyE	Median (range)	19.1 (0.0–37.6)	18.0 (0.0–35.4)	19.2 (0.0–37.6)	0.909 ^‡^
_Bowel_D_2cc_, GyE	Median (range)	11.5 (0.0–36.3)	4.3 (0.0–36.3)	12.3 (0.0–35.1)	0.289 ^‡^
_Cord_D_2cc_, GyE	Median (range)	11.9 (0.0–29.6)	8.7 (0.0–29.6)	13.3 (0.0–19.9)	0.992 ^‡^

Abbreviations: ECOG PS, Eastern Cooperative Oncology Group performance status; CA 19–9, carbohydrate antigen 19–9; ND, newly diagnosed; IHR, intrahepatic recurrence; Tx, treatment; PBT, proton beam therapy; CTx, chemotherapy; EQD2, equivalent dose in 2 Gy fractions (EQD2 = total dose × [(fraction dose + α/β)/(2 + α/β)], α/β = 10); GyE, gray equivalent (GyE = proton physical dose [in gray] × relative biological effectiveness [1.1]); TL, total liver; RRL, remaining residual liver; _TL_V_27GyE_, relative volume of the TL receiving ≥27 GyE; _RRL_V_27GyE_, relative volume of the RLL receiving ≥27 GyE; D_2cc_, delivered radiation dose to the stomach, esophagus, duodenum, bowel, and spinal cord of 2 cc (cm^3^). * Maximum tumor diameter ^†^ Fisher’s exact test. ^‡^
*t*-test. ^§^ Capecitabine (n = 5), 5-fluorouracil (n = 4), gemcitabine, and cisplatin (n = 1)

**Table 2 cancers-14-05561-t002:** Univariate analysis of the pre-treatment characteristics for the free from local progression (FFLP), progression-free survival (PFS), and overall survival (OS).

			FFLP		PFS		OS	
Characteristics		N	2 yr (95% CI), %	*p*-Value *	2 yr (95% CI), %	*p*-Value *	2 yr (95% CI), %	*p*-Value *
Sex	Male	32	92.3 (82.1–102.5)	0.521	22.1 (7.2–37.0)	0.555	48.9 (27.8–70.0)	0.919
	Female	15	75.0 (44.0–106.0)		17.9 (−4.2–40.0)		33.3 (9.4–57.2)	
Age, years	<70	28	84.2 (67.5–100.9)	0.633	21.2 (5.3–37.1)	0.632	51.7 (32.5–70.9)	0.019
	≥70	19	93.3 (80.8–105.8)		17.2 (−3.4–37.8)		29.2 (8.0–50.4)	
ECOG PS	0	28	81.8 (65.3–98.3)	0.180	26.0 (9.5–42.5)	0.100	55.7 (38.9–74.5)	<0.001
	1–2	19	100 (–)		9.8 (9.5–42.5)		23.0 (2.4–43.6)	
Hilum invasion	No	34	91.5 (80.1–102.9)	0.172	19.0 (3.9–34.1)	0.818	48.1 (30.7–65.5)	0.648
	Yes	13	76.4 (46.0–106.8)		23.1 (0.2–46.0)		30.8 (5.7–55.9)	
Tumor size, cm	<5	21	84.7 (64.5–104.9)	0.604	33.8 (12.6–55.0)	0.190	50.8 (28.8–72.8)	0.251
	≥5	26	90.0 (76.9–103.1)		9.4 (−2.9–21.7)		36.3 (17.1–55.5)	
T classification	T1-2	9	100 (–)	0.133	40.0 (5.9–74.1)	0.048	88.9 (68.3–109.5)	0.027
	T3-4	38	81.6 (64.0–99.2)		15.6 (3.3–27.9)		32.6 (17.3–47.9)	
N classification	N0	36	93.2 (84.0–102.4) ^†^	0.593	53.3 (36.4–70.2)	0.001	50.5 (33.6–67.4)	0.001
	N+	11	85.7 (59.8–111.6) ^†^		10.0 (−8.6–28.6)		18.2 (−4.5–40.9)	
M classification	M0	36	93.3 (84.3–102.3) ^†^	0.521	53.2 (36.3–70.1)	< 0.001	50.5 (33.6–67.4)	< 0.001
	M1	11	83.3 (53.5–113.1) ^†^		10.0 (−8.6–28.6)		18.2 (−4.5–40.9)	
Stage	I/II	8	100 (–) ^†^	0.327	75.0 (45.0–105.0)	< 0.001	100 (–)	<0.001
	III	28	90.9 (78.7–103.1) ^†^		46.5 (29.5–63.5)		37.7 (19.5–55.9)	
	IV	11	83.3 (53.5–113.1) ^†^		10.0 (−7.7–27.8)		18.2 (−0.4–36.8)	
CA 19–9, U/mL	<100	27	89.1 (74.6–103.6)	0.670	31.1(12.1–50.1)	0.084	57.4 (38.0–76.8)	0.003
	≥100	20	85.9 (67.7–104.1)		5.7 (−5.1–16.5)		22.5 (3.5–41.5)	
Biliary drainage	No	40	89.7 (78.7–100.7)	0.862	21.7 (7.6–35.8)	0.864	48.3 (32.2–64.4)	0.395
	Yes	7	66.7 (13.4–120.0)		14.3 (−11.6–40.2)		14.3 (−11.6–40.2)	
Disease status	ND	38	87.3 (73.2–101.4)	0.968	17.1 (4.2–30.0)	0.378	43.3 (27.2–59.4)	0.450
	IHR	9	85.7 (59.8–111.6)		33.9 (−1.8–69.6)		40.0 (6.7–73.3)	
Pre-Tx to PBT site(s)	No	26	95.2 (85.8–104.6)	0.286	29.7 (10.9–48.5)	0.010	55.9 (36.1–75.7)	0.116
	Yes	21	74.6 (47.7–101.5)		6.3 (−5.5–18.1)		27.2 (7.6–46.8)	
Concurrent CTx	No	37	84.7 (70.4–99.0)	0.380	23.8 (8.5–39.1)	0.511	49.4 (32.7–66.1)	0.044
	Yes	10	100 (–)		10.0 (−8.6–28.6)		20.0 (−4.7–44.7)	
Post-Tx after PBT	No	23	90.0 (71.4–108.6)	0.556	35.2 (13.1–57.3)	0.108	51.5 (30.9–72.1)	0.418
	Yes	24	83.6 (66.5–100.7)		8.7 (−2.9–20.3)		34.4 (14.4–54.4)	
Total dose	<90	24	77.1 (52.8–101.4)	0.474	13.7 (−0.9–28.3)	0.226	25.0 (7.4–44.2)	0.147
(EQD2, GyE_10_)	≥90	23	94.7 (84.7–104.7)		27.0 (7.7–47.0)		59.9 (39.5–80.3)	
	<80	12	66.7 (25.9–107.5)	0.303	16.7 (−4.5–37.9)	0.357	25.0 (0.5–49.5)	0.215
	≥80	35	92.7 (82.9–102.5)		21.0 (5.7–36.3)		49.5 (32.4–66.6)	
Viable tumor(s)	Absence	29	86.7 (72.4–101.0)	0.992	33.2 (14.6–51.8)	<0.001	63.3 (45.1–81.5)	< 0.001
outside of PBT site(s)	Presence	18	90.9 (73.8–108.0)		0.0 (–)		8.3 (−6.0–22.6)	

Abbreviations: CI, confidence interval; ECOG PS, Eastern Cooperative Oncology Group performance status; CA 19–9, carbohydrate antigen 19–9; Tx, treatment; PBT, proton beam therapy; CTx, chemotherapy; EQD2, equivalent dose in 2 Gy fractions (EQD2 = total dose × [(fraction dose + α/β)/(2 + α/β)], α/β = 10); GyE, gray equivalent (GyE = proton physical dose [in gray] × relative biological effectiveness [1.1]); * log-rank test. ^†^ 1 year.

**Table 3 cancers-14-05561-t003:** Multivariate analysis of the pre-treatment characteristics for the free from local progression (FFLP), progression-free survival (PFS), and overall survival (OS).

		FFLP		PFS		OS	
Characteristics		HR (95% CI)	*p*-Value *	HR (95% CI)	*p*-Value *	HR (95% CI)	*p*-Value *
ECOG PS	0	-	-	-	-	1.000	0.014
	1–2	-		-		2.631 (1.212–5.712)	
CA 19–9, U/mL	<100	-	-	-	-	1.000	0.006
	≥100	-		-		2.883 (1.365–6.091)	
Viable tumor(s)	Absense	-	-	1.000	<0.001	1.000	<0.001
outside of PBT site(s)	Presence	-		6.007 (2.686–13.432)		5.891 (2.577–13.471)	

HR, hazard ratio; ECOG PS, Eastern Cooperative Oncology Group performance status; CA 19–9, carbohydrate antigen 19–9; PBT, proton beam therapy; CI, confidence interval. * Cox proportional hazards model.

**Table 4 cancers-14-05561-t004:** Adverse events related with proton beam therapy.

	All Patients (n = 47)			
CTCAE Grade	Grade 1, n (%)	Grade 2, n (%)	Grade 3, n (%)	Grade 4, n (%)
Hematologic AEs	20 (42.6)	6 (12.8)	4 (8.5)	0 (0.0)
WBC increase	6 (12.8)	0 (0.0)	0 (0.0)	0 (0.0)
WBC decrease	7 (14.9)	5 (10.6)	2 (4.3)	0 (0.0)
Hb decrease	7 (14.9)	2 (4.3)	0 (0.0)	0 (0.0)
PLT decrease	4 (8.5)	2 (4.3)	0 (0.0)	0 (0.0)
ALT/AST increase	7 (14.9)	0 (0.0)	0 (0.0)	0 (0.0)
Albumin decrease	4 (8.5)	0 (0.0)	0 (0.0)	0 (0.0)
Bilirubin increase	6 (12.8)	0 (0.0)	2 (4.3)	0 (0.0)
Non-hematologic AEs	19 (40.4)	3 (6.4)	0 (0.0)	0 (0.0)
Fever	1 (2.1)	0 (0.0)	0 (0.0)	0 (0.0)
Pain	1 (2.1)	2 (4.3)	0 (0.0)	0 (0.0)
Dermatitis	15 (31.9)	3 (6.4)	0 (0.0)	0 (0.0)
Radiation pneumonitis	12 (25.5)	0 (0.0)	0 (0.0)	0 (0.0)
Upper gastrointestinal ulcer	0 (4.3)	1 (2.1)	0 (0.0)	0 (0.0)

Abbreviations: CTCAE, Common Terminology Criteria for Adverse Events, (version 5.0); n, number of patients; WBC, white blood cell; Hb, hemoglobin; PLT, platelet; ALT, alanine aminotransferase; AST, aspartate aminotransferase.

## Data Availability

All datasets of the present study are available upon formal request from the corresponding author.

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
