# Peer review of "Clinical Efficacy of Hypofractionated Proton Beam Therapy for Intrahepatic Cholangiocarcinoma"

_cancers, 2022, doi:10.3390/cancers14225561_

Round 1

Reviewer 1 Report

General Comments

This is a retrospective review of 47 patients with intrahepatic cholongiocarcinoma (IHCC) treated over a 9 year period with hypofractionated proton beam therapy (PBT). The authors found that local control was excellent and the toxicity of PBT was relatively low.  IHCC is a rare disease, and this series does contribute to the literature on treatment.  The authors recognize that the patient population is quite heterogeneous at presentation and in the treatments received, and they do not over interrupt their data.

Specific Comments

1)    Pg 2 line 89: How was the tumor defined?  IHCCs are difficult to image on contrast CT.  Were gadolinium-enhanced MRIs used?

2)    Pg 3 para line 117: Was cone beam CT used for imaging?  (I would assume that is what was meant by “daily imaging”.

3)    What were the criteria for giving radiation therapy?  In particular, what was the goal of local treatment in patients with metastatic disease treated? 

4)    What were the criteria for receiving concurrent and post treatment chemotherapy?

5)    It would be interesting to see a typical radiation treatment plan.

6)    It is somewhat surprising that patients who received concurrent chemoradiation had a worse survival that those who did not. Do the authors have an explanation for this?  This factor did not appear in the multi-variate analysis so there are likely other factors involved.

7)    Could the authors discuss more about why they chose hypofractionated treatment and why they think that this approach might be superior to stereotactic body radiation therapy.  There is still significant discussion about the higher cost of PBT compared to photon SBRT from a standard linear accelerator, so this discussion might put this paper in better perspective for a general audience.

Author Response

Point by Point Responses to Reviewer 1’ comments

This is a retrospective review of 47 patients with intrahepatic cholongiocarcinoma (IHCC) treated over a 9 year period with hypofractionated proton beam therapy (PBT). The authors found that local control was excellent and the toxicity of PBT was relatively low.  IHCC is a rare disease, and this series does contribute to the literature on treatment.  The authors recognize that the patient population is quite heterogeneous at presentation and in the treatments received, and they do not over interrupt their data. 

Specific Comments

1)    Pg 2 line 89: How was the tumor defined?  IHCCs are difficult to image on contrast CT.  Were gadolinium-enhanced MRIs used?

à Response: We appreciate you for your kind considerations and insightful comments on our paper. According to reviewer’s comments, we revised the manuscript as follow,

In Page 2 line 49 – Page 3 line 6,

“The gross tumor volume (GTV) was defined in average intensity projection CT images at exhalation (gated) phases (i.e., 30% of the entire respiration cycle in each patient) based on available imaging studies including liver dynamic CT and/or MRI. The internal target volume (ITV) and organs at risk (OARs) were defined to account for the extent and movement of the GTV and each OAR at the exhalation (gated) phases and the clinical target volume was defined as the ITV with no additional margins”

2)    Pg 3 para line 117: Was cone beam CT used for imaging?  (I would assume that is what was meant by “daily imaging”.

à Response: According to reviewer’s comments, we revised the manuscript as follow,

In Page 3 line 32,

“… during the gated phases after localization using digital orthogonal and/or cone beam CT images.”

3)    What were the criteria for giving radiation therapy?  In particular, what was the goal of local treatment in patients with metastatic disease treated? 

à Response: The use of local treatment including PBT for intrahepatic disease was decided by multidisciplinary evaluation by medical, surgical, and radiation oncologists depending on extent of intrahepatic and/or extrahepatic disease and each patient’ performance status and underlying medical conditions. According to reviewer’s comments, we revised the manuscript as follow,

In Page 2 line 28 – 36,

“All the patients underwent multidisciplinary evaluation by medical, surgical, and radiation oncologists for assessment of resectability and selection of treatment modalities, i.e., the use, sequence and regimens of systemic treatments, and the use of local treatments including PBT considering the extent of disease and each patient’s performance status (PS) and underlying medical conditions and were staged … Eastern Cooperative Oncology Group (ECOG) PS, tumor size and stage,…”

4)    What were the criteria for receiving concurrent and post treatment chemotherapy?

à Response: The use of concurrent and post treatment chemotherapy was decided by physicians considering each patient’s performance status and age. We revised the manuscript as follow,

In Page 4 line 20 – 26,

“…(Table S1). During PBT, concurrent chemotherapy was considered in 39 patients, except for 8 patients with stage I/II, but 10 (25.6%) patients received concurrent chemotherapy by the physicians’ decision considering the patient’s PS and age (Table 1). The median EQD2 of PBT was 80 GyE10 (range, 54.4–120.0):35 (74.5%) patients received >80 GyE10, and 12 (25.5%) patients received ≤ 80 GyE10. After PBT, subsequent treatments (including chemotherapy) were considered in all the patients based on their PS and age, and 24 (51.1%)…”

5)    It would be interesting to see a typical radiation treatment plan.

à Response: According to reviewer’s comments, we added the supplementary Figure S1 showing a typical radiation treatment plan and imaging feature of IHCC after PBT. Numbering of pre-exiting Supplementary figures was changed (i.e., S1 àS2, S2 à S3)

In Page 3 line 8 – 11,

“PBT … covered at least 95% of the PTV (Figure S1).

In Page 6 line 7,

“…not reached (Figure S2A).”

In Page 6 line 11,

“… 2 and 3) (Figure S3A), and there .”

In Page 8 line 16,

“…months) (Figure S2B).”

In Page 8 line 22,

“…(p>0.05 each) (Table 2) (Figure S3B).”

In Page 8 line 26,

“… months) (Figure S2C).”

In Page 8 line 33,

“…(Table 2) (Figure S3C).”

In Page 11 line 32 – 38,

“Figure S1: An example of proton beam therapy (PBT) for intrahepatic cholangiocarcinoma (IHCC). (A) Computed tomography (CT) scans at diagnosis showing the tumor (arrow). (B) Plan CT scans for proton beam therapy (PBT) showing dose line. (C and D) CT scans at 6 and 12 months, respectively, after PBT showing marked shrinkage of the tumor (arrow). Figure S2: Freedom from local progression (FFLP) (A), progression-free survival (PFS) (B), and overall survival (OS) (C) curves in patients with intrahepatic cholangiocarcinoma treated with proton beam therapy; Figure S3: Freedom from local progression (FFLP) (A),…”

6)    It is somewhat surprising that patients who received concurrent chemoradiation had a worse survival that those who did not. Do the authors have an explanation for this? This factor did not appear in the multi-variate analysis so there are likely other factors involved.

à Response: The patients who did not receive current chemotherapy was included the patients with stage I/II (n=8), but the patients who received concurrent chemotherapy included only stage III-IV. Thus, chemotherapy did not significant in multivariate analysis. We revised the manuscript as follow,

In Page 4 line 20 – 26,

“…(Table S1). During PBT, concurrent chemotherapy was considered in 39 patients, except for 8 patients with stage I/II, but 10 (25.6%) patients received concurrent chemotherapy by the physicians’ decision considering the patient’s PS and age (Table 1).”

7)    Could the authors discuss more about why they chose hypofractionated treatment and why they think that this approach might be superior to stereotactic body radiation therapy. There is still significant discussion about the higher cost of PBT compared to photon SBRT from a standard linear accelerator, so this discussion might put this paper in better perspective for a general audience.

à Response: According to reviewer’s comments, we added the summary of studies for patients with IHCC treated with various RT techniques including local control, survival, and adverse event (Supplementary Table S2). We revised the manuscript as follow,

In Page 11 line 27 – 28,

“Although the need for high-cost dedicated equipment for PBT has limited clinical application of PBT for patients with IHCC to date, further prospective large-scale studies…”

In Page 11 line 40 – 41,

“…proton beam therapy (PBT); Table S2. Summary of studies for intrahepatic cholangiocarcinoma treated with radiotherapy.”

Reviewer 2 Report

This study deals with the novel application of hypofractionated proton beam therapy intrahepatic cholangiocarcinoma. This paper is clear and concise and describes well the clinical features of the 47 patients treated. However, this paper would gain in impact if the following points  could be addressed : 

- in introduction, a table with references of comparable studies and notably the irradiation protocols  should be added in order to reply to the question : what is the origin of the anti-tumor efficiency : the use of protons or the hypofractionation of the dose? This question can be also discussed in the discussion chapter.

- in materials and methods : the formulas for the calculation of GyE and EQD2 should be detailed

- in discussion the distribution of CTCAE grades is an important feature. It would be important to compare it with those from other comparables studies. 

Some typos to be corrected : ex: first sentence of the Discussion chapter

An interesting study that needs to be better illustrated and discussed with some tables summarizing comparable studies.

Author Response

Point by Point Response to Reviewer 2’s comments

This study deals with the novel application of hypofractionated proton beam therapy intrahepatic cholangiocarcinoma. This paper is clear and concise and describes well the clinical features of the 47 patients treated. However, this paper would gain in impact if the following points could be addressed : 

- in introduction, a table with references of comparable studies and notably the irradiation protocols should be added in order to reply to the question : what is the origin of the anti-tumor efficiency : the use of protons or the hypofractionation of the dose? This question can be also discussed in the discussion chapter.

--> Response: We appreciate you for your kind considerations and insightful comments on our paper. According to reviewer’s comments, we added the supplementary table summarizing comparable studies showing dose protocols and radiotherapy techniques (i.e., conventional RT, SBRT, IMRT, PBT, etc.) (Supplementary Table S2).

In Page11 line 40 – 41,

“…proton beam therapy (PBT); Table S2. Summary of studies for intrahepatic cholangiocarcinoma treated with radiotherapy.”

- in materials and methods : the formulas for the calculation of GyE and EQD2 should be detailed

--> Response: According to reviewer’s comments, we revised the manuscript as follow,

In Page 3 line 12 – 15,

“…Gray equivalent (GyE = physical dose of proton beam [Gray] × relative biologic equivalent factor of proton beam [1.1]) and was converted to the equivalent dose for 2 Gy fractions (EQD2 [GyE3 or GyE10] = [(fraction dose+ α/β)/(2+α/β)]×total dose) with α/β values of 10 (acute responding tissues and tumor) or 3 (late responding tissues) (GyE10 or GyE3, respectively)…”

- in discussion the distribution of CTCAE grades is an important feature. It would be important to compare it with those from other comparables studies. 

--> Response: According to reviewer’s comments, we added the supplementary table summarizing comparable studies comparing clinical outcomes including adverse events (Supplementary Table S2).

In Page 11 line 40 – 41,

“…proton beam therapy (PBT); Table S2. Summary of studies for intrahepatic cholangiocarcinoma treated with radiotherapy.”

Some typos to be corrected : ex: first sentence of the Discussion chapter

-->Response: According to reviewer’s comments, we revised the manuscript as follow,

In Page 9 line 14,

“For patients with inoperable or recurrent IHCC, no standard treatment option for liver-directed local treatments has been established…”

An interesting study that needs to be better illustrated and discussed with some tables summarizing comparable studies.

--> Response: According to reviewer’s comments, we added the supplementary table summarizing comparable studies (Supplementary Table S2).

In Page 11 line 40 – 41,

“…proton beam therapy (PBT); Table S2. Summary of studies for intrahepatic cholangiocarcinoma treated with radiotherapy.”
